# IWIN: The Isfjorden Weather Information Network

Lukas Frank[1,2], Marius Opsanger Jonassen[1,2], Teresa Remes[3], Florina Roana Schalamon[1,4,a], and Agnes Stenlund[1,5,b]

[1]Department of Arctic Geophysics, University Centre in Svalbard, Longyearbyen, Norway
[2]Geophysical Institute, University of Bergen, Bergen, Norway
[3]Development Centre for Weather Forecasting, Norwegian Meteorological Institute, Oslo, Norway
[4]Institute for Atmospheric Physics, Johannes Gutenberg University, Mainz, Germany
[5]Department of Earth Sciences, Uppsala University, Uppsala, Sweden
[a]now at: Department of Geography and Regional Sciences, University of Graz, Graz, Austria
[b]now at: Department of Environmental Science, Stockholm University, Stockholm, Sweden

**Correspondence:** Lukas Frank (lukasf@unis.no)

**Abstract.** In an effort led by the University Centre in Svalbard (UNIS), with support from the Norwegian Meteorological Institute (MET Norway), the Isfjorden Weather Information Network (IWIN) is under development in the Isfjorden region, central Svalbard. The network substantially expands upon the relatively sparse existing operational network of weather stations and consists of compact and cost-efficient all-in-one weather stations permanently installed on lighthouses around Isfjorden and onboard small tourist cruise ships trafficking the fjord from spring to autumn. All data from the network are freely available online in near real-time via MET Norway's data portals (https://doi.org/10.21343/ebrw-w846). The IWIN data are highly valuable for scientific purposes such as atmospheric boundary layer research, validation and development of numerical weather prediction models and assimilation in these, as well as planning and safe conduct of outdoor activities in the region.

## 1 Introduction

In this study, we present the Isfjorden Weather Information Network (IWIN), which is a new network of automatic weather stations located in the Isfjorden area, central Svalbard. The network is developed by the University Centre in Svalbard (UNIS) with support from the Norwegian Meteorological Institute (MET Norway).

IWIN consists of compact and relatively cost-efficient all-in-one weather stations measuring near-surface temperature, humidity, wind speed, wind direction and pressure. The stations are robust with no movable parts and thereby well-suited for the harsh Arctic climate in Svalbard. The stations are mounted at both fixed points (lighthouses) situated along the shoreline of Isfjorden and on small tourist ships that traffic the fjord from spring to autumn. Hence, the network uses existing infrastructure as instrument platforms and its (added) environmental footprint is therefore minimal. IWIN is under continuous development and as of summer 2023 it consists of 7 weather stations, 4 of which are mounted on lighthouses and 3 of which are mounted on ships (in the following referred to as mobile stations). The data from IWIN are made freely and publicly available in near real-time on MET Norway's THREDDS server (https://thredds.met.no/thredds/unis-obs/unis-obs.html) and via the Arctic Data Centre (ADC, https://doi.org/10.21343/ebrw-w846).

The usefulness of the IWIN observations is multifold. From a research perspective, IWIN provides valuable in-situ, near-surface weather observations from the Arctic, where such observations are otherwise very sparse. The network supports our need to better document and understand the ongoing strong climate warming in Svalbard (Isaksen et al., 2022), which is well beyond the pan-Arctic warming rate of nearly 4 times the global average (Rantanen et al., 2022). Embedded in the Svalbard climate change are effects such as sea ice retreat (Muckenhuber et al., 2016; Dahlke et al., 2020), extreme precipitation events (Müller et al., 2022) and rain-on-snow events (Peeters et al., 2019; Wickström et al., 2020). Future climate projections indicate further warming and more of these climate change related effects in the decades to come (Hanssen-Bauer et al., 2019, e.g.). The complex topography of Svalbard exacerbates the need for more observations from the archipelago, as the weather typically varies substantially in space due among others to flow phenomena such as gap winds (Jackson and Steyn, 1994), channelling effects (Skeie and Gronas, 2000, e.g.) and katabatic winds (Esau and Repina, 2012). With its rugged coast line, surrounded by steep mountain ranges and deep valleys, the Isfjorden area is no exception. The atmospheric component (and its related forcing) of the fjord system furthermore acts as a boundary condition for geological, physical and biological interactions in the region (Cottier et al., 2007; Walczowski and Piechura, 2011; Nilsen et al., 2016; Descamps et al., 2017; Skogseth et al., 2020; Schuler et al., 2020).

Numerical models are integral parts of several of the above-cited studies of weather and climate processes in Svalbard. However, such models typically struggle to accurately represent weather and climate processes in the Arctic (Jung et al., 2016). Recent progress has yielded promising results (Bromwich et al., 2016), and MET Norway's operational Arome Arctic model (covering Svalbard and northern Fennoscandia, Müller et al. (2017)) has been shown to perform favourably compared to e.g. the global forecasting model of the European Centre of Medium-Range Weather Forecasting (Køltzow et al. (2019)). Further progress in model development, both related to validation and assimilation purposes, relies heavily on more observations such as those provided by IWIN. High-resolution observations are especially useful as we progress towards hectometric scale model simulations, which are currently in testing for Svalbard (Valkonen et al., 2020).

IWIN data are also useful in a societal context, especially because the Isfjorden region is the most populated area in Svalbard. Here, we find the settlements of Longyearbyen, Barentsburg and Pyramiden, and human activity is widespread in the form of, among others, fishery, tourism and research activities. In particular the two latter peak during summertime. Furthermore, emergency situations may occur at any time of the year in the harsh arctic environment of Svalbard, frequently sparking search and rescue missions. Online, near real-time weather observations, like those provided by IWIN, are key to keeping the planning and conduction of such activity as safe as possible.

The primary goal of this paper is to provide documentation on the instrumental setup of IWIN and describe and evaluate the data it produces. This includes introductions to the observation locations and information on the instrumentation in use (Section 2) as well as a description of the data handling process (Section 3). In Section 4, we provide an evaluation of the quality of the data set and discuss remaining uncertainties. By presenting several examples of weather phenomena observed through IWIN in Section 5, we draw connections between these and the data evaluation as well as highlight the novel capabilities of the network and indicate potential usage for future work. In the end, we summarize the current status of IWIN in Section 6 and give an outlook on the further development of the network in Section 7.

## 2 Weather Station Network

Comprising a combination of stationary and mobile automatic weather stations, IWIN provides near-surface observations of temperature (T), relative humidity (RH), pressure (p), wind speed (WS) and wind direction (WD) from a large portion of
Isfjorden. Thus, IWIN complements the long-term reference surface weather stations operated by MET Norway and located around the fjord at Isfjord Radio (IR), Svalbard Airport (SA), Nedre Sassendalen (NS) and Pyramiden (PYR). As some of the land-based IWIN station locations are inaccessible for large parts of the year, using rugged all-in-one weather stations with no moving parts ensures low risk of failure and long maintenance intervals. By making use of already existing infrastructure, observations from remote areas are obtained at low cost, while at the same time the local, additional environmental impact of
these observations can be considered negligible. All observations are automatically transferred from the stations to UNIS in regular intervals via the 4G cellular network. In the following, the individual station locations are introduced in more detail.

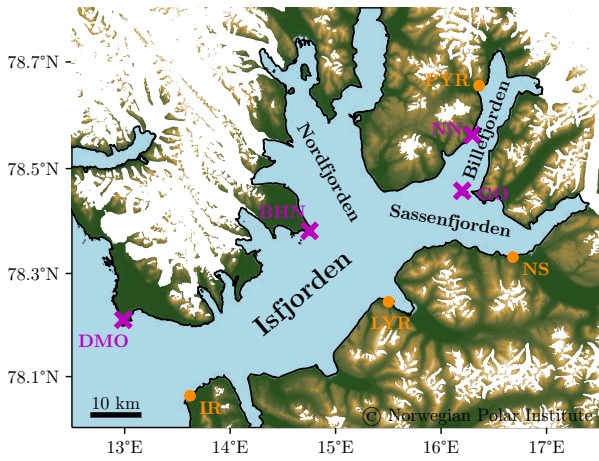

**Figure 1.** Overview map of the Isfjorden area. Local fjord names are given in black, locations of MET Norway weather stations are marked with orange dots and the IWIN lighthouse stations are marked with magenta crosses. See main text and Table 1 for full station names. This and all following map figures are produced using map data from the Norwegian Polar Institute (https://geodata.npolar.no).

### 2.1 Lighthouse Stations

Presently, IWIN comprises four stationary stations around Isfjorden (for the exact geographical locations see Table 1 and Figure 1). The stations are installed on top of small coastal lighthouses (see Figure 2), approximately 3.6 m above ground level. The
instruments used are Campbell Scientific MetSens500 sensors, configured to measure T, RH, p, WS and WD at a raw sampling frequency of 5 seconds. While T, RH and p are measured by solid state sensor circuits located inside a radiation shield in the lower part of the instrument, a 2D-sonic anemometer on top provides WS and WD.

**Table 1.** Overview of lighthouse stations. The sensors are installed on top of the lighthouses, approximately 3.6 m above ground level.

| Location (abbreviation) | Latitude [°N] | Longitude [°E] | Measurement altitude (above sea level) [m] | Installation date |
| --- | --- | --- | --- | --- |
| Bohemanneset (BHN) | 78.38166 | 14.75300 | 12 | 19 August 2021 |
| Narveneset (NN) | 78.56343 | 16.29687 | 7 | 17 June 2022 |
| Daudmannsodden (DMO) | 78.21056 | 12.98685 | 39 | 08 July 2022 |
| Gåsøyane (GØ) | 78.45792 | 16.20082 | 19 | 03 September 2022 |

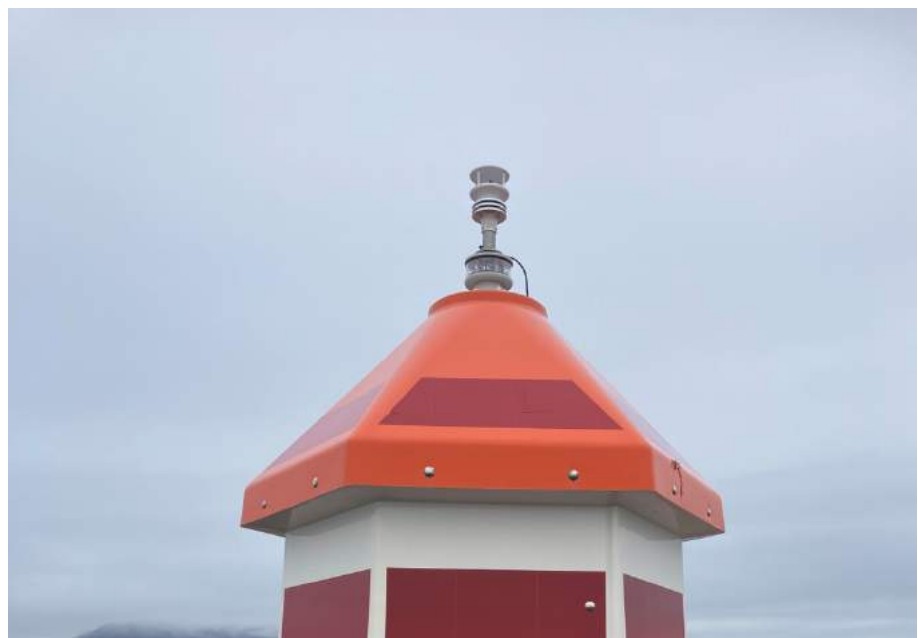

**Figure 2.** Weather station installed on top of the Gåsøyane lighthouse.

The first lighthouse station was installed at Bohemanneset (BHN) in August 2021. This lighthouse is located at 8 m above sea level on the tip of the flat headland Bohemanflya. The location sticks out approximately 10 km into central Isfjorden in the south-easterly direction (see Figure 1), in relation to the nearest major mountains in the northwest. The wind rose from this station (Figure 3) shows the prevailing wind directions to be aligned with three major (side-)fjord axes. The main sectors NW–N and NE–E resemble outflow from Nordfjorden and Sassenfjorden, respectively. The secondary peak for SW directions matches inflow into Isfjorden along the main fjord axis. The highest wind speeds almost exclusively occur with flow out of Sassenfjorden (see the example case presented in Section 5 for more details and the importance of channeling effects on the local near-surface winds there).

After the BHN station had operated flawlessly throughout winter 2021/2022, three more lighthouse stations were installed during the summer of 2022. The first of these (installed in mid-June 2022) is located at the western shoreline in central

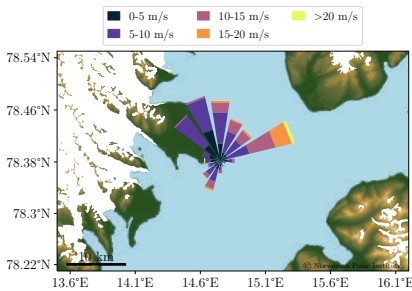

**Figure 3.** Wind climatology at Bohemanneset, including all data available at the time of publication (19 August 2021 – 22 June 2023).

Billefjorden, on a small headland called Narveneset (NN, approximately 3 m above sea level, see Figure 1). Billefjorden is the innermost sidearm of Isfjorden and is surrounded by steep topography. As could be expected, the wind observations from NN show a very dominant alignment with the fjord axis (NE–SW, see Figure 4).

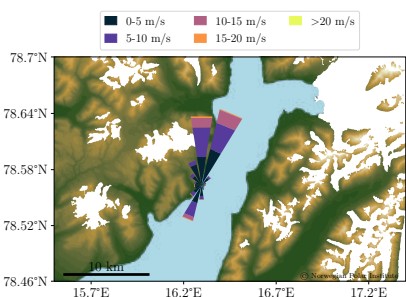

**Figure 4.** Wind climatology at Narveneset, including all data available at the time of publication (17 June 2022 – 22 June 2023).


 The station at Daudmannsodden (DMO, installed in the beginning of July 2022) is situated at the northern shore right outside the mouth of Isfjorden (see Figure 1). The location is distanced from the nearest coastal mountain ranges by the approximately 8 km wide Daudmannsøyra headland. The lighthouse itself is located on a small hill rising approximately 20 m above the surrounding flatland, resulting in a total station height of 35 m above sea level. The observations from this station exhibit
90 stronger maritime characteristics than those from the other stations (e.g. smaller annual temperature range and on average higher specific humidity, see also evaluation results in Section 4), due to the influence of the large open water body in Fram Strait and the warm water masses transported northwards there by the West Spitsbergen Current. Winds from westerly sectors are fairly equally distributed over all respective directions, presumably due to no guiding topography upstream of the station for those sectors (see wind rose in Figure 5). The main (easterly) peak in the wind rose is still strongly related to outflow
95 through the mouth of Isfjorden and winds aligned with the valleys in the nearby coastal mountain ranges.

 The hitherto last IWIN lighthouse station was installed at Gåsøyane (GØ) in the beginning of September 2022. Gåsøyane is a group of small islands at the intersection of Billefjorden and inner Sassenfjorden (see Figure 1). The lighthouse with the

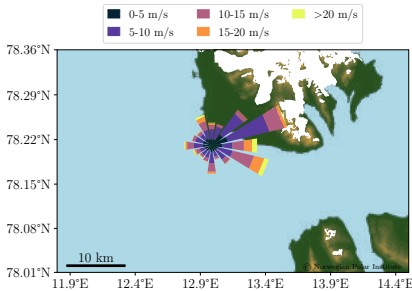

**Figure 5.** Wind climatology at Daudmannsodden, including all data available at the time of publication (08 July 2022 – 22 June 2023).

station on top is located on the largest of these small islands, on a small cliff approximately 15 m above sea level. The axes of both Billefjorden and inner Sassenfjorden (NE–SW and SE–NW) are strongly imprinted in the observed wind direction distribution (see wind rose in Figure 6). Additionally, a secondary peak resembles inflow into outer Sassenfjorden.

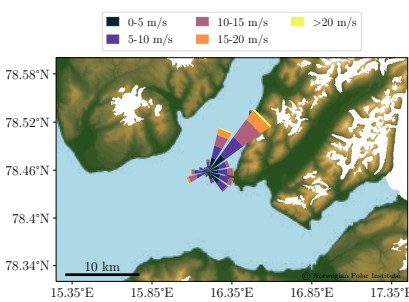

**Figure 6.** Wind climatology at Gåsøyane, including all data available at the time of publication (02 September 2022 – 22 June 2023).


    As of June 2023, all lighthouse stations have been measuring continuously since their respective installation dates (see Table 1 and Figure 7). In combination with MET Norway's reference surface weather stations in the region, they provide a good overview of the weather conditions in different parts of the Isfjorden system at all times. Furthermore, different subsets of the network can for example be used to investigate local gradients within the fjord system, e.g. along the main fjord axis (DMO/IR

– BHN/SA – GØ – NN – PYR) or across the fjord (DMO – IR, BHN – SA, NN – GØ – NS).

## 2.2 Mobile Stations

In addition to the lighthouse stations operating year-round, a set of mobile weather stations complete the IWIN network from spring to autumn each year (see Figure 7). During the first season of operation in 2021, two stations were installed on the tourist cruise ships MS Bard and MS Polargirl. The former is a modern, 24 m long catamaran with a hybrid-electric propulsion

system. In 2023 MS Bard was renamed MS Berg and this ship will hereafter be referred to as MS Bard/Berg. MS Polargirl is a 35 m long passenger vessel with a classic streamlined design. In 2022, a third station was added on MS Billefjord, which

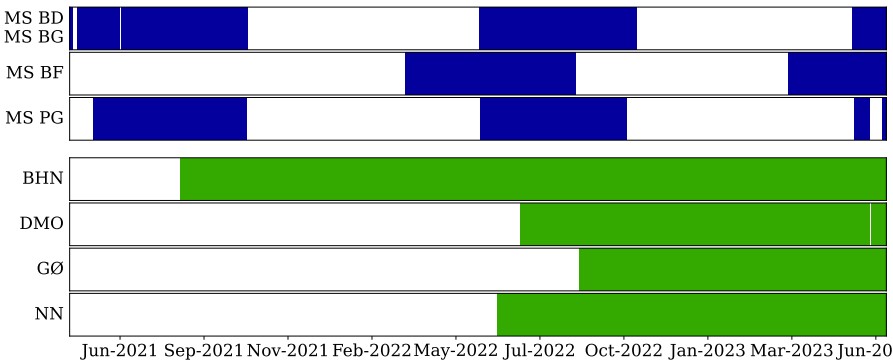

**Figure 7.** Temporal data availability from the installation of the first station in spring 2021 until 22 June 2023. The mobile stations are colored in blue, the lighthouse stations in green.

has similar dimensions and design as MS Polargirl. From day to day, the three ships follow different schedules and routes, however, they generally visit the same destinations including mainly Pyramiden, Barentsburg and the glaciers in the north- and eastern parts of Isfjorden. Combining the measurements from all mobile stations results in the track pattern and corresponding

spatial data density indicated in Figure 8. One can see that the tracks cover large parts of Isfjorden from the mouth area in the west to the head of Billefjorden in the east.

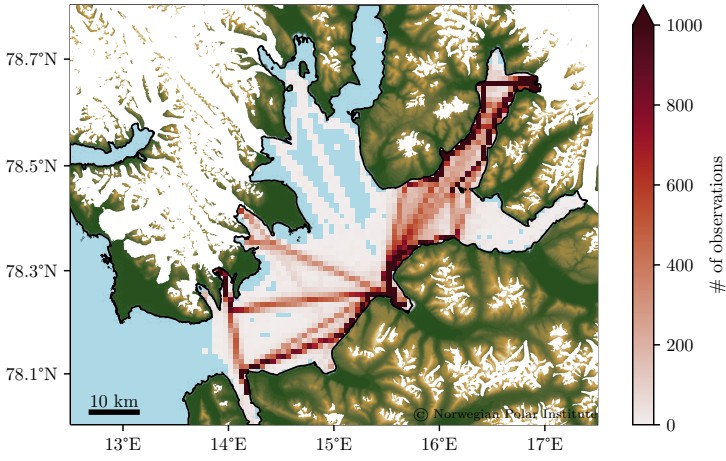

**Figure 8.** Overview map of the tracks from the ships MS Bard/Berg, MS Polargirl and MS Billefjord where observations are available from the 3 mobile weather stations from the seasons 2021, 2022 and 2023 (until 22 June). The spatial density of observations is discretized (counted) in boxes of size approximately 1x1 km and indicated in shades of red, based on a temporal data resolution of 1 min.

The mobile stations are installed at the top of the ships' masts at 19 – 21 m above sea level (see Figure 9), to minimize disturbance of the measurements caused by flow distortion from the ships' own structures. Furthermore, sea spray rarely reaches that high up, which reduces the risk of icing in the beginning and the end of the season, when temperatures often fall below zero. The sensors used are Gill MaxiMet GMX500, configured to sample at a raw frequency of 1 Hz. The individual parameters are measured in the same way as with the lighthouse stations (T, RH, p: solid state sensor circuits; WS, WD: sonic anemometer). In addition to the standard meteorological variables, they provide heading and GPS position measurements.

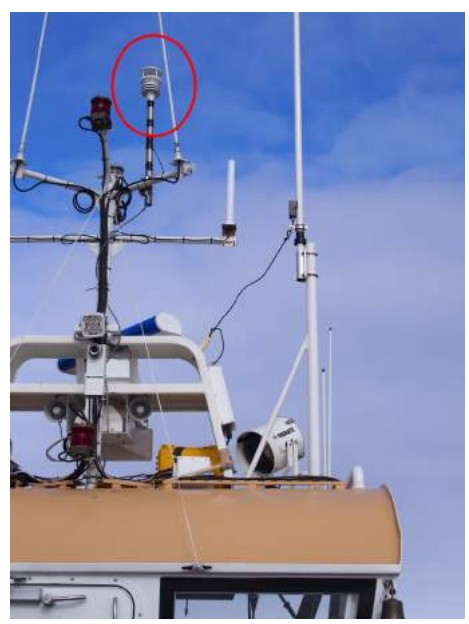

**Figure 9.** Weather station installed on top of MS Bard/Berg, marked with a red circle.

## 3   Data Processing and Storage

The raw measurements undergo a series of processing steps before being published online. This starts with the averaging of the raw data over 1 min and 10 min intervals directly in the data logger connected to the individual weather stations. The data measured by the mobile stations additionally get averaged over 20 s intervals, for a higher spatial resolution of the data along the ship tracks.[1]

After the initial averaging, the data are transferred to UNIS via the 4G cellular network at intervals of 2 and 30 min for the mobile and lighthouse stations, respectively. The differing transfer intervals are chosen because of power consumption considerations, especially with regards to the data modems that consume relatively much power when they are active. While the mobile stations are connected to the ships' onboard power supply and power consumption is not a limitation, the lighthouse

---

[1]Prior to May 2022, T, RH and p from the mobile stations and T from the lighthouse station at BHN were sampled as instantaneous values at the end of each averaging interval. This was changed to the above-described averaging procedure in order to unify the IWIN dataset.

stations run on the lighthouses' battery banks. These are charged via solar panels over summer, but have to bridge the polar night (approximately end of October until end of February).

At UNIS, the meteorological data first get filtered for nonphysical outliers by discarding data outside predefined valid ranges
for each variable (specified in the final data files). The mobile station data are additionally filtered for erroneous GPS positions located outside the Isfjorden area. Subsequently, the raw wind measurements from the mobile stations are corrected for the ships' horizontal movements using the GPS speed and heading data via basic vector geometry during times when the ships are moving faster than 0.25 m/s (approx. 0.5 knts). Based on the raw wind direction measurements relative to the ships, an additional variable is introduced to mark data points potentially affected by the ship's exhaust plume (see Table 2 for the
respective wind direction sectors). Note that MS Bard/Berg does not have this problem as its exhaust funnel is located down at sea level.

The final data are stored in daily files using the netCDF4 format (individual files for each temporal resolution). This data format allows for extensive meta data to be added directly to each file. For the measured meteorological variables, this includes, amongst others, information about the valid ranges applied in the filtering (see above), the physical units of the measurements
and standardized naming. Furthermore, each data file includes a set of global attributes with a summary of its contents as well as information about the instrument to measure the raw data, the data processing steps and contact information of the creators. The data files are conform with the Climate and Forecast Metadata Convention (CF-Convention, version 1.8), and the Attribute Convention for Data Discovery (ADCC, version 1.3), which makes them machine-readable and easy to access.

**Table 2.** Overview of mobile stations and respective exhaust plume sectors.

|  | exhaust plume sector [1] |
| --- | --- |
| MS Bard/MS Berg | no filtering needed |
| MS Polargirl | $215 - 235\,°$ |
| MS Billefjord | $170 - 190\,°$ |

[1] relative to the ship, $0\,°$ resembles head wind

The final data files are transferred from UNIS every 5 minutes via Secure File Transfer Protocol to a virtual server owned by
MET Norway. From there, the data become publicly available in near real-time (time lag of approximately $5 - 10$ min) at their Thematic Real-time Environmental Distributed Data Services- (THREDDS) server (https://thredds.met.no/thredds/unis-obs/unis-obs.html) as well as at the Arctic Data Centre (ADC, https://doi.org/10.21343/ebrw-w846, Frank et al. (2023)).

## 4 Measurement Evaluation and Remaining Uncertainties

The harsh Arctic environment of Svalbard presents challenges to equipment placed in the field for continuous measurements
over long time periods. The instrumentation used in IWIN has been chosen accordingly. With no movable parts, the sensors are robust and well suited for operation under a wide range of environmental conditions. Prior to installation, all sensors were

factory-calibrated and quality-checked with respect to the measurement specifications stated by the manufacturers (see Table 3). Even though the reference temperatures accompanying these specifications are rather high compared to the measurements in Isfjorden, the specified operating range of the sensors (-40 °C – +70 °C) is large enough to cover even the coldest temperature events occurring in the region.

**Table 3.** Measurement resolutions and accuracies for the sensors used in IWIN as stated by the manufacturers (lighthouse stations: https://s.campbellsci.com/documents/us/product-brochures/b_metsens500.pdf, mobile stations: https://gillinstruments.com/wp-content/uploads/2022/08/1957-008-Maximet-gmx500-Iss-9.pdf).

|  | T | RH | p | WS | WD |
|---|---|---|---|---|---|
| resolution | 0.1 K | 1 % | 0.1 hPa | 0.01 m/s | 1 ° |
| accuracy | ± 0.3 K at 20 °C | ± 2 % at 20 °C | ± 0.5 hPa at 25 °C | ± 3 % up to 40 m/s | ± 3 % up to 40 m/s |


## 4.1 Evaluation of Lighthouse Station Data

Given their remote locations, validation of the lighthouse station measurements against an absolute reference is difficult. However, MET Norway's surface reference stations at Isfjord Radio, Svalbard Airport and Nedre Sassendalen (see Figure 1) can be said to constitute a small ensemble representing the general weather conditions in the Isfjorden area. Figure 10 shows the

measurement time series for the period 02 September 2022 – 22 June 2023 (period since the so far last lighthouse station in Gåsøyane became operational) from all three MET Norway reference stations and all four IWIN lighthouse stations. It can be seen that all seven stations agree qualitatively very well and exhibit the same overall temporal variations. Furthermore, it can be seen that measurements of individual variables tend to have reoccurring deviations at certain locations, e.g. lower temperature in Nedre Sassendalen during winter and higher wind speed at Daudmannsodden. Table 4 corroborates these findings,

summarizing the corresponding, quantitative comparison statistics for the lighthouse stations using the ensemble mean of the MET Norway data as a reference.

Daudmannsodden in the west is the warmest location with an average deviation from the ensemble mean (bias) of 0.37 K. Narveneset in the east, in contrast, is the coldest location with a (marginal) bias of -0.02 K. Indeed, Gjelten et al. (2016) found a corresponding east-west gradient in their study of spatial temperature variability in the Isfjorden region, attributing the warmer

temperatures in the west to a stronger influence of the open ocean in Fram Strait. The lighthouse stations are on average biased moist (positive relative humidity biases) and the strongest bias of 8.33 % is found at Bohemanneset. It is unclear what would cause these moist biases, but the lighthouses are all located very close to the Isfjorden shoreline, which might make them more influenced by moisture from the sea compared to the (mean of the) MET Norway stations. Another explanation might be a slight calibration offset at the lighthouse stations. Air pressure is generally a little lower at the lighthouse stations (negative

biases) and the largest average deviation (bias) of -0.60 hPa is found at Daudmannsodden. The pressure MAEs are of the order of 0.5 hPa (maximum 0.91 hPa at Daudmannsodden). Wind speed is on average higher at the lighthouse stations, with the only exception being Narveneset with a bias of -1.02 m/s. It is likely that larger portions of the wind speed biases can be attributed

to natural atmospheric variability in the wind field, for example induced by topography. Indeed, the wind climatologies (wind roses) presented in Section 2.1 as well as the last example presented in Section 5 feature considerable, topography-induced, variability.

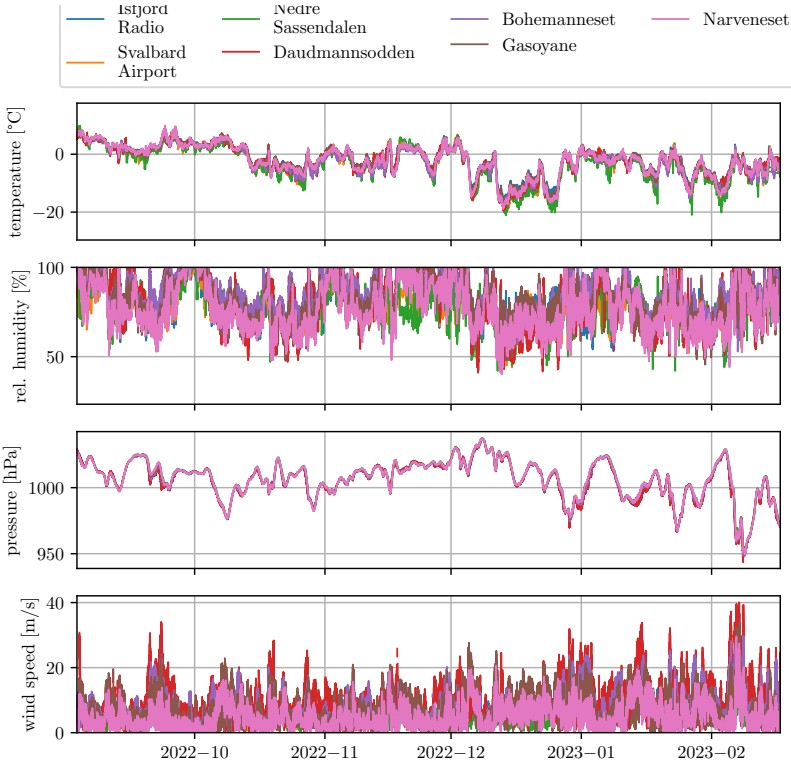

**Figure 10.** Time series of T, WS, RH and p from the four IWIN lighthouse stations and three additional MET Norway stations around Isfjorden. The pressure measurements from the individual stations have been reduced to mean sea level to account for the differences in station elevation.

## 4.2 Evaluation of Mobile Station Data

Evaluation of the IWIN mobile station data has been done through three sets of comparisons as outlined in the following.

### 4.2.1 Internal Comparison of Mobile Station Data

The first comparison set addresses the consistency in data across the different mobile stations for two subsets of situations when the three respective ships were located in the near vicinity of each other: 1) when all three ships were docked in the Longyearbyen harbor (78.2285°N, 15.607°E) and 2) when pairs of ships passed each other (distance < 1 km) during transit on Isfjorden. The former subset does not include wind direction as the estimation of this variable relies on the ships moving

**Table 4.** Lighthouse station comparison statistics for T, RH, p and WS, calculated with respect to the ensemble mean of the three MET Norway surface reference stations at Isfjord Radio, Svalbard Airport and Nedre Sassendalen.

|  |  | T [K] | RH [%] | p [hPa] | WS [m/s] |
|---|---|---|---|---|---|
| Bohemanneset | Bias | 0.01 | 8.33 | -0.49 | 1.43 |
|  | MAE | 1.04 | 8.88 | 0.56 | 2.54 |
| Narveneset | Bias | -0.02 | 0.83 | -0.07 | -1.02 |
|  | MAE | 1.04 | 6.41 | 0.51 | 2.80 |
| Daudmannsodden | Bias | 0.37 | 5.86 | -0.60 | 2.92 |
|  | MAE | 1.14 | 9.22 | 0.91 | 3.87 |
| Gåsøyane | Bias | 0.22 | 4.80 | -0.18 | 1.10 |
|  | MAE | 0.93 | 7.02 | 0.52 | 3.25 |

and the latter subset has been filtered for GPS speed values lower than 0.25 m/s to ensure reliable wind direction estimates. The results from this first set of comparisons are presented in Table 5. For both subsets (Longyearbyen harbor and Isfjorden) the temperature and relative humidity comparison statistics (bias and MAE) are within the accuracies given by the instrument manufacturer (Table 3) and hence indicate excellent matches between the three mobile stations for these variables. These results also suggest that the influence of any sources of deviations between the stations are minimal, such as calibration offsets, the different ships' physical characteristics (their size, shape etc.). The pressure measurements also agree well between the different stations for both subsets and most numbers are within the given factory accuracy. One smaller exception is the station on MS Polargirl, which on average yields slightly higher pressure values than the other two stations (e.g. 0.78 hPa higher pressure on average than MS Billefjord during transit on Isfjorden). This is likely caused by a slight calibration offset of the pressure sensor on this station. Wind speed compares favourably between the stations for the Longyearbyen harbor subset and, even though MAE values are higher than the manufacturer's accuracy, the values are still fairly low with the highest MAE of 0.57 m/s found between MS Bard/Berg and MS Polargirl. The corresponding wind speed comparison statistics for the Isfjorden subset are somewhat less favourable than for the Longyearbyen harbor subset, with the highest MAE value of 1.11 m/s found between MS Bard/Berg and MS Billefjord. Given the distance of up to 1 km between the ships on Isfjorden it is likely that natural atmospheric variability in wind might be a substantial source of deviation between the data sets. As illustrated in the wind climatology from the lighthouse stations (Section 2.1) and in the last example in Section 5 below, considerable spatial variability in wind over Isfjorden has indeed been documented by the mobile stations, even over distances shorter than 1 km. Wind direction displays the largest difference between the mobile stations. Note that the sign-convention of the wind direction bias is such that positive is clock-wise. Although the biases are relatively low (within 12.53 °), the MAEs reach 30.05 ° (comparing MS Bard/Berg and MS Polargirl). The ships' differing physical characteristics as well as natural variability are both likely sources of deviations, but there might also be further sources involved. One such source is the alignment of the

sensors with the ships. It is generally challenging to align the stations north/south axis exactly along the ships' "x-axis" (from bow to stern) when mounting them on the ships. It is likely that slight offsets related to this alignment are a source of deviation. Another source of deviation is the uncertainty in the GPS heading estimates during times the ships move very slowly. Even though a threshold of 0.25 m/s for the ship speeds is used in the correction algorithm, it can not be completely ruled out that uncertainty in the GPS heading estimates impacts the corrected wind direction observations during times a ship moves only slightly faster than the threshold speed. When using wind direction data from the mobile IWIN stations it is important to take these uncertainties in the wind direction measurements into account and apply additional filters or corrections if needed.

**Table 5.** Mobile station comparison statistics for T, RH, p, WS and WD for times when all three ships were docked in the Longyearbyen harbor (LH) and for times when they were passing each other during transit on Isfjorden within a distance of less than 1 km (IS). 1 min data are used for the comparison and the number of data points (n) are given in hours.

| | | T [K] | | RH [%] | | p [hPa] | | WS [m/s] | | WD [°] | | n [hrs] | |
| | | LH | IS | LH | IS | LH | IS | LH | IS | LH | IS | LH | IS |
|---|---|---|---|---|---|---|---|---|---|---|---|---|---|
| MS Bard/Berg vs MS Polargirl | Bias | 0.06 | 0.02 | -0.09 | 0.20 | -0.26 | -0.48 | 0.25 | 0.46 | N/A | 5.82 | 1182 | 9 |
| | MAE | 0.09 | 0.19 | 0.95 | 1.64 | 0.27 | 0.52 | 0.57 | 1.13 | N/A | 29.91 | | |
| MS Bard/Berg vs MS Billefjord | Bias | -0.08 | -0.14 | -0.08 | 0.02 | 0.32 | 0.07 | 0.07 | 0.08 | N/A | 4.19 | 1182 | 6 |
| | MAE | 0.15 | 0.26 | 0.8 | 1.55 | 0.36 | 0.52 | 0.5 | 1.11 | N/A | 28.79 | | |
| MS Polargirl vs MS Billefjord | Bias | -0.13 | -0.16 | 0.01 | 0.37 | 0.57 | 0.78 | -0.18 | -0.34 | N/A | 12.51 | 1182 | 17 |
| | MAE | 0.15 | 0.22 | 1.01 | 1.24 | 0.59 | 0.81 | 0.45 | 0.89 | N/A | 21.29 | | |

### 4.2.2 Comparison of Mobile Station and Lighthouse Station Data

The second comparison set contrasts the measurements between the three mobile stations and the two lighthouse stations at Narveneset and Gåsøyane for time steps when the mobile stations are within a distance of 2 km from the respective lighthouse station. The temperature comparison statistics for Gåsøyane are all below or only slightly above the sensor accuracy of 0.3 K (Table 6). The Narveneset statistics are slightly worse and all mobile station data are on average slightly colder compared to this lighthouse station (-0.35 K for MS Polargirl). A closer investigation of these numbers reveals that this cold bias is stronger during summer (June-July) than autumn (September-October). For example, comparing MS Bard/Berg and MS Polargirl to Narveneset yields -0.46 and -0.53 K during summer and -0.04 and -0.22 K during autumn. This is likely an effect of the air over land being heated more than the air over sea during summer. The Gåsøyane comparison does not display a similar tendency, which can likely be explained by this location being situated on a smaller island that is more exposed to the (air from the) sea. In terms of relative humidity, the comparison statistics are fairly favourable and only slightly beyond the manufacturers' accuracy. A notable exception are the statistics for the MS Billefjord and Gåsøyane comparison where there is a bias of -4.92 % and an MAE of 4.98 %. However, this comparison is based on data from only a short time period in spring 2023 (see limited

overlap in data availability from the two stations in Figure 7). The pressure data comparison statistics are very favourable and mostly within the manufacturers' accuracy of 0.5 hPa. An exception is found for MS Polargirl, with both bias and MAE values reaching 1.35 hPa when compared against Narveneset. This fits with the previously suggested calibration offset for pressure measured by the mobile station on MS Polargirl. The wind speed measured by the three mobile stations is on average higher than that at Narveneset, and for Gåsøyane the opposite is the case. This could be linked to the measurement heights above sea level. While both lighthouse stations are located at about 3.6 m above the ground level, their altitudes (base of lighthouses) above sea levels differ with Narveneset being located at about 3 m above sea level and Gåsøyane at 15 m above sea level. This is presumably particularly relevant for wind directions at Gåsøyane coming from a northern and northeastern sector, where there is a cliff in the immediate vicinity (30-50 m distance) of the lighthouse. This is in contrast to southern and southwestern sectors where this lighthouse faces relatively flat, open land for several hundred meters. Stratifying the comparison statistics by wind direction measured at Gåsøyane indeed reveals differences. While wind from between 20 and 60$^\circ$ (north-northeast) at Gåsøyane gives biases of -1.60, -1.87 and -1.41 m/s for MS Bard/Berg, MS Polargirl and MS Billefjord, respectively, the corresponding biases for wind from between 180 and 290° (south-west) are 0.27, 0.19 and -0.09 m/s. It is likely that north-northeast wind is locally accelerated over the cliff at Gåsøyane and that this at least in part explains the statistics' sensitivity to wind direction at this location. The wind direction statistics are quite similar between the different mobile and lighthouse stations. Also, at least in terms of MAE, they are similar to the comparison between the mobile stations on the different ships (Section 4.2.1) with values between 20 and 30$^\circ$.

**Table 6.** Mobile station comparison statistics for T, RH, p, WS and WD calculated with respect to the lighthouse stations at Narveneset (NN) and Gåsøyane (GØ) for times when the ships were within 2 km of these stations. 1 min data are used for the comparisons and the number of data points (n) are given in hours.

|  |  | T [K] | | RH [%] | | p [hPa] | | WS [m/s] | | WD [°] | | n [hrs] | |
|---|---|---|---|---|---|---|---|---|---|---|---|---|---|
|  |  | NN | GØ | NN | GØ | NN | GØ | NN | GØ | NN | GØ | NN | GØ |
| MS Bard/Berg | Bias | -0.30 | 0.06 | 0.12 | -1.98 | 0.4 | 0.31 | 0.79 | -0.12 | -4.77 | -7.71 | 20 | 8 |
|  | MAE | 0.41 | 0.31 | 2.43 | 3.04 | 0.54 | 0.47 | 1.31 | 1.56 | 23.08 | 28.03 | | |
| MS Polargirl | Bias | -0.35 | 0.18 | 0.95 | -2.36 | 1.35 | 0.97 | 0.00 | -0.58 | -11.29 | -13.72 | 17 | 4 |
|  | MAE | 0.45 | 0.31 | 2.92 | 3.34 | 1.35 | 0.97 | 0.9 | 1.49 | 22.85 | 29.97 | | |
| MS Billefjord | Bias | -0.17 | 0.08 | -1.20 | -4.92 | 0.37 | 0.12 | 0.59 | -0.80 | -10.24 | 19.00 | 27 | 8 |
|  | MAE | 0.48 | 0.32 | 2.87 | 4.98 | 0.41 | 0.29 | 1.06 | 1.57 | 27.91 | 23.49 | | |

### 4.2.3 Comparison of Mobile Station Data with Reference Station Data

The third comparison set consists of two subsets contrasting mobile station data with 1) data from the MET Norway reference station at Svalbard Airport (SA) for times when the ships were within 2 km of this station and 2) data from the Norwegian

research vessel Kronprins Haakon (KH) for times when this ship was docked in the Longyearbyen harbor alongside the ships with the mobile stations. For the comparison against SA, only data when the ships were moving (higher GPS speed than 0.25 m/s) have been considered. Also, mobile station data from east of 15.529°E have been excluded from the comparison because an industrial site with several buildings is located between this longitude and SA. In addition, for the SA wind data comparison, only cases when the wind at SA originated from a sector between 220 and 300 ° (roughly southwest to northwest) have been considered for the same reason.

Regarding measurement heights at SA and KH, the former follows the conventional 2 and 10 m heights while at KH temperature, relative humidity and pressure are measured at 21 m above sea level and wind speed and wind direction are measured at 36 m above sea level. Compared against the SA measurements, the mobile station data are biased cold on MS Bard/Berg and MS Polargirl (-0.38 and -0.24 K) and slightly warm on MS Billefjord (0.19 K). Stratifying these data by summer (June-July) and autumn (September-October) for the two former ships reveals consistent negative biases during summer (-0.98 and -0.78 K) and positive biases during autumn (0.50 and 0.65 K). MS Billefjord has not operated during autumn, but the spring (March-April) data reveals a positive bias of 1.53 K, versus a summer data bias of -0.67 K. Given the seasonal dependency on the sign of these biases, it is likely that they feature large contributions from natural atmospheric variability. Indeed, the air above land is generally warmer than over the sea during summer in the Isfjorden region and vice versa during spring and autumn. Compared against the measurements from KH, all mobile station data yield small temperature biases between 0.05 K at MS Billefjord and -0.10 K at MS Polargirl. The corresponding MAEs are also relatively small between 0.41 K (MS Bard/Berg) and 0.53 K (MS Billefjord). The mobile station data are all biased moist against both the SA and the KH measurements, with a bias of up to 6.76 % for MS Bard/Berg compared to SA. Considering only the SA comparison, one could expect the mobile station data to be moister as these were measured over the sea and SA measured over land. However, the KH comparison gives similar moist biases and these data were obtained from virtually the same location as the mobile station data. It is not clear what causes these moist biases in the mobile station measurements. However, as similar moist biases have also been found for the lighthouse stations, the particular humidity sensors in the Gill Maximet GMX 500 and Campbell Scientific MetSens500 instruments used in IWIN could be biased somewhat moist in general.

The pressure data error statistics are the least favourable for MS Polargirl yielding positive biases of 0.79 and 1.06 hPa against SA and KH. This is in line with the previously found tendency for this mobile station to report slightly higher pressure values than the other mobile stations. The error statistics for MS Billefjord are all within the manufacturers' stated accuracy of 0.5 hPa, while for MS Bard/Berg the values are slightly larger with an MAE of up to 0.93 hPa when compared against KH. The wind speed error statistics reveal very small biases for all three mobile stations, especially compared to SA but also compared to KH. The latter biases are consistently negative, which fits with the wind measurements at KH being obtained at 31 m above sea level compared to 19 – 21 m above sea level for the mobile stations. In terms of MAE, the values are least favourable for the comparison against SA, ranging between 1.00 m/s (MS Billefjord) and 1.66 m/s (MS Bard/Berg). This is not surprising given the differing underlying surfaces (land vs sea) and horizontal distances (up to 2 km) involved, and it is reasonable to assume that natural atmospheric variability plays a role in these deviations. Regarding wind direction, the measurements from

MS Billefjord feature the least favourable error statistics, with an MAE of 36.26 ° when compared against SA. It is again clear that wind direction is the variable from the mobile stations with the lowest accuracy.

**Table 7.** Mobile station comparison statistics for T, RH, p, WS and WD, calculated with respect to 1) measurements from the MET Norway reference station at Svalbard Airport (SA) for times when the ships were within 2 km of this station and 2) measurements from the Norwegian research vessel Kronprins Haakon (KH) when this was docked in the Longyearbyen harbor alongside the three ships. The number of data points (n) are also indicated. Note that SA reports T, RH and p every hour (SA[1]) and WS and WD every 10 min (SA[2]). KH reports all variables every hour.

| | | T [K] | | RH [%] | | p [hPa] | | WS [m/s] | | WD [°] | | n | | |
| --- | --- | --- | --- | --- | --- | --- | --- | --- | --- | --- | --- | --- | --- | --- |
| | | SA | KH | SA | KH | SA | KH | SA | KH | SA | KH | SA[1] | SA[2] | KH |
| MS Bard/Berg | Bias | -0.38 | -0.07 | 6.76 | 4.90 | -0.04 | 0.80 | -0.18 | -0.25 | -3.58 | N/A | 58 | 144 | 39 |
| | MAE | 0.76 | 0.44 | 7.14 | 4.90 | 0.51 | 0.93 | 1.66 | 0.45 | 32.73 | N/A | | | |
| MS Polargirl | Bias | -0.24 | -0.10 | 6.51 | 5.69 | 0.76 | 1.06 | -0.16 | -0.20 | -19.65 | N/A | 39 | 81 | 39 |
| | MAE | 0.71 | 0.41 | 6.92 | 5.69 | 0.81 | 1.18 | 1.09 | 0.35 | 28.71 | N/A | | | |
| MS Billefjord | Bias | 0.19 | 0.05 | 3.09 | 4.82 | -0.05 | 0.33 | 0.17 | -0.42 | -29.52 | N/A | 32 | 66 | 39 |
| | MAE | 0.92 | 0.53 | 4.03 | 4.82 | 0.39 | 0.46 | 1.00 | 0.58 | 36.26 | N/A | | | |

### 4.2.4 Remaining Uncertainties

In summary, the evaluation presented above documents the quality of the IWIN dataset and underlines its potential for investigations on the temporal and spatial atmospheric variability in the Isfjorden region.

Some sources of uncertainty remain, however, especially with regards to the mobile stations and their wind direction data, which display the largest inaccuracies in comparison to each other and the other stations. As discussed above, there are likely several sources for these inaccuracies, including the sensors' alignment with the ships, the ships' differing physical characteristics (their shape, size etc), uncertainties in the ships' heading estimates from GPS data and not least natural atmospheric variability when comparing the mobile stations to references located on land and in a distance of up to 2 km. Efforts are currently underway to improve the accuracy of the IWIN mobile stations' wind measurements. New and better routines for aligning the sensors (north-south axes) with the ships' bow-stern axes during the annual installation process will be investigated. In addition, a GPS-based compass for better estimation of the ships' heading is currently under testing with MS Berg. The results from this are still under evaluation and, if proven successful, this compass should also allow wind direction measurements when the ships are standing still (as opposed to the now minimum GPS speed of 0.25 m/s). Furthermore, a recent master study at UNIS carried out CFD simulations for investigating flow distortion around MS Bard/Berg (Reen, 2022). The results from this study are still to be formally published and a thorough analysis is beyond the scope of this paper, but they do indicate quantitatively little influence of this flow distortion on the wind measurements on that ship at the location of the sensors (at the ship's highest point). The generally excellent match between the different ships' wind speed measurements (Ta-

ble 5) support this finding, as the three ships' physical characteristics are rather different. The same applies to the temperature and relative humidity measurements, which also match excellently between the ships. Besides the actual horizontal motion of the ships, pitching and rolling motions in heavy seas introduce additional artificial wind components measured by the stations.

However, the temporal averaging of the raw data acts as a low-pass filter for this high-frequency variability and their impact on the final data set is thus presumably small. Finally, with sub-zero temperatures for extended periods during a year, icing and/or riming might cause sensor malfunctioning, especially for the (non-heated) exposed sonic anemometer transducer heads. As the ships used as platforms for the mobile stations do not operate during winter and the sensors itself are mounted high up on the masts, the danger of icing/riming can be considered rather small. The lighthouse stations are mounted in remote locations

virtually inaccessible during conditions when icing and/or riming might occur. Therefore, we have not been able to visually monitor them for such issues. However, from the data records we do not see any indications of icing and/or riming impacting the measurements.

## 5 Natural Atmospheric Variability: Examples of Observed Weather Phenomena and Linkages to the Evaluation of the IWIN Data

As discussed in the evaluation of the IWIN observations (Section 4), natural atmospheric (spatial) variability has an impact on the comparison statistics and at least in part explains offsets between the measurements from the various IWIN and reference weather stations. The inferred atmospheric variability includes among others (seasonally dependent) contrasts between near-surface temperatures over land and sea, as seen for example in the comparisons of the mobile station data against data from Svalbard Airport and Narveneset. In addition, there are signs of impacts from topographic effects on the wind field, as deduced

from the comparison between the mobile station data and data from Gåsøyane. In this section, examples will be given of such atmospheric variability and connections will be made to the aforementioned evaluation of the IWIN data. Furthermore, the examples demonstrate the capabilities of IWIN to observe local weather phenomena and show potential for future use of the data set. A further, in-depth analysis of the weather phenomena is, however, beyond the scope of this paper.

The first example is taken from 20 October 2022. On this day, Svalbard was under the influence of a high-pressure ridge

extending over the western central Arctic and Greenland, and a low pressure system centered over Novaya Zemlya. This setup established a weak but well-defined northerly synoptic flow over Svalbard, advecting relatively cold Arctic air masses over the Isfjorden region. Figure 11 shows near-surface wind and temperature data this day from Billefjorden and the adjacent Isfjorden proper, as observed by the mobile station on MS Bard/Berg. Both the local temperature as well as wind field display rather strong variability. The lowest temperatures are generally found close to the entrance of valleys and several of these valleys

have wind emanating from them, affecting areas such as close to Pyramiden, Kapp Ekholm, Rundodden, Skansbukta and Narveneset. Given the low temperatures at these locations, the relatively weak synoptic flow and the local wind directions (along the valley axes), the air masses are likely driven by a thermal component (drainage flow), set up by the land-sea temperature contrast, bringing a terrestrial atmospheric (in this case cold) footprint onto the fjord. Other locations, such as along the shoreline between Narveneset and Pyramiden, and indeed also the stretch of Billefjorden covered by MS Bard/Berg

as it passed by Gåsøyane, do not show signs of such a terrestrial footprint. MS Bard/Berg's close proximity to the lighthouses Narveneset and Gåsøyane this day allows a detailed comparison of the IWIN data from these sources, and a brief analysis of how the described natural atmospheric variability affects this. Figure 12 shows time series of the data from MS Bard/Berg and the lighthouses at Narveneset and Gåsøyane, measured during the 20-minute periods when the ship passed by the respective lighthouse. The red shading highlights the sub-period corresponding to the 2-km-threshold used in the calculation of the

comparison statistics in Section 4.2.2. One can see that small-scale spatial variability like e.g. the drop in temperature observed by MS Bard/Berg when passing the valley entrance south of the Narveneset lighthouse (at about 08:57) is not captured by the stationary measurements at the lighthouse. Comparing the magnitude of this temperature drop (approx. 1 K) with the results from the mobile station evaluation presented in Table 6 (MAEs approx. $0.3 - 0.5$ K) exemplifies how natural spatial variability in the near-surface atmospheric variables can contribute to large parts of the differences between the stations. Similarly, the

wind speed at Narveneset featured substantially less variability for this comparison than that recorded by the mobile station on MS Bard/Berg, contributing to differences of up to 5 m/s (at about 09:01). In the comparison of data from MS Bard/Berg and Gåsøyane, the recorded temperatures match very well, both being dominated by a fetch from Billefjorden (northerly flow). The wind speed, however, is higher at Gåsøyane than at MS Bard/Berg and this can likely be related to the aforementioned topographic effect (acceleration of the wind) at Gåsøyane for northerly flow, as discussed in Section 4.2.2.

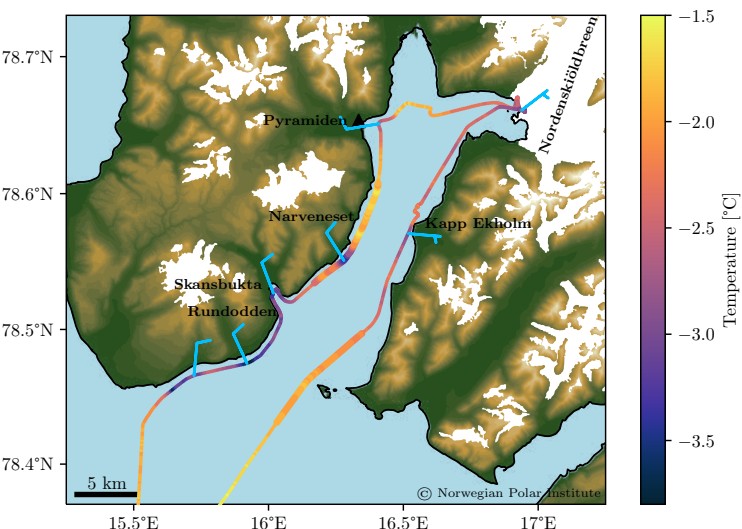

**Figure 11.** Temperature (solid line, colored shading) and winds (wind barbs) over Isfjorden, as observed by the IWIN mobile weather station on MS Bard/Berg during the time 07:00 – 13:00 UTC on 20 October 2022. Those parts of the ship track presented with increased marker size correspond to the data comparison time series shown in Figure 12.

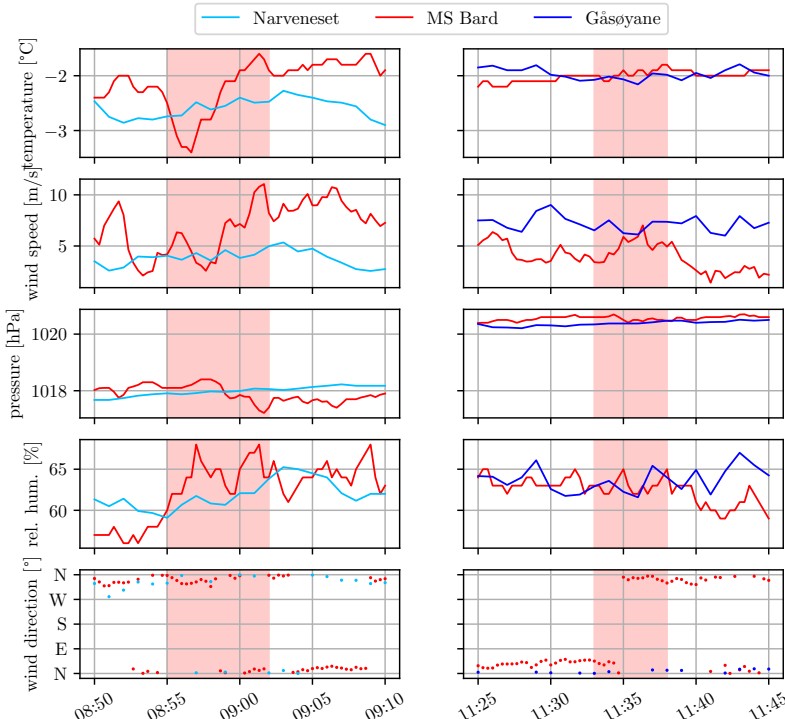

**Figure 12.** Comparison of time series measured onboard MS Bard/Berg and at the two lighthouses at Narveneset (left column) and Gåsøyane (right column) on 20 October 2022 for the time periods the ship passed by the respective lighthouse. The respective part of the ship track is indicated with an increased marker size in Figure 11. The data that contributed to the calculation of the comparison statistics presented in Table 6 in Section 4 are highlighted with a light-red background color.

The second example is from spring 2022, covering the period 04 – 06 April. The example demonstrates marine and terrestrial footprints on the air masses by the Bohemanneset lighthouse station, which is located right at the shoreline of Isfjorden and therefore ideally situated to measure both types of air masses. During the days in question, Svalbard was situated under a weak to moderately strong easterly flow field, set up by a high-pressure system to the north and a low pressure system to the south. However, locally at Bohemanneset (Figure 13 (a)), the wind direction varied, resulting in several large step changes in temperature and specific humidity. For instance, there was a marked drop in both temperature and specific humidity in the first few hours of 04 April by almost 8 K and 0.5 g/kg, followed by an equally sharp increase just minutes later. These step changes coincide in time with a change in wind direction from NE to NW and back to NW. A closer look at the satellite picture in Figure 13 (b) reveals that air masses advected from NE originated from over the open water (marine footprint) and air masses advected from NW originated from land (terrestrial footprint) at Bohemanneset. It stands to reason that the relatively warm and moist air mass characteristics over the fjord are related to the impact of the heat and moisture release from the fjord surface. In

contrast, air masses advected from the terrestrial sector (NW) have been cooled and dried over land. The comparison of IWIN mobile station data with land-based station data from Svalbard Airport and Narveneset is likely impacted by corresponding effects from natural, spatial atmospheric variability.

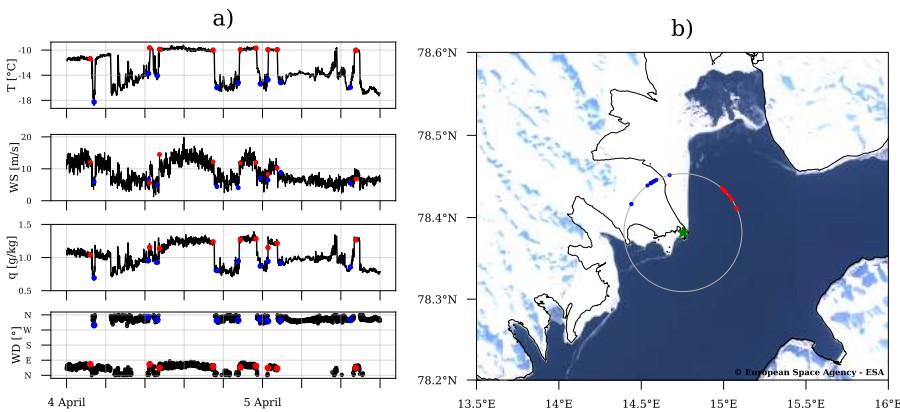

**Figure 13.** (a) Example time series from the IWIN weather station at Bohemanneset from 04 – 06 April 2022 of temperature (T), wind speed (WS), specific humidity (q) and wind direction (WD) during a period when changes in WD induce step-changes in the other three parameters. (b) True color satellite image (composite of two Sentinel-2 overpasses on 05 and 11 April 2022, produced from European Space Agency (ESA) remote sensing data, downloaded from the Copernicus Hub, https://scihub.copernicus.eu/dhus/) centred on the location of Bohemanneset (green star). The coast line is marked by a black, solid line. Notable positive and negative step-changes are indicated with respectively red and blue markers in both the time series of T and WD (a) and in the map on a circle centred on Bohemanneset (b).

The third example is from 5 August 2022 and uses mobile station data from IWIN to showcase horizontal extent of terrestrial
temperature footprint on airmasses over Isfjorden. During this day the large-scale synoptic forcing was weak over Svalbard. Neither low-pressure systems situated over the Norwegian Sea and the Kara Sea nor a high-pressure ridge in the central Arctic extended far enough towards Svalbard to noticeably impact the weather conditions in the region resulting in weak wind conditions over Isfjorden. For the period of the ship tracks from MS Bard/Berg and MS Polargirl shown in Figure 14 (06:00 – 17:00 UTC), the average wind speed measured at Bohemanneset was 2.3 m/s and came from N–E directions (not shown),
indicating predominantly marine influence at this location. In the absence of strong synoptic-scale forcing, combined with throughout overcast conditions and polar day, local factors related to the landmasses surrounding Isfjorden can be expected to dominate the near-surface atmospheric conditions experienced over different parts of the fjord. Figure 14 shows the anomaly in observed temperature on MS Bard/Berg and MS Polargirl, using the temperature at Bohemanneset as reference (lighthouse minus mobile station data). A marked gradient is evident in the near-surface temperature anomaly field over Isfjorden, when
going from the relatively warm southeastern shore and central parts of the fjord (about 8.5 – 9.5 °C) to the relatively cold northernwestern shore (about 6.5 – 7.5 °), especially towards the two bays on that side of the fjord. This temperature gradient is the opposite of what is otherwise found during summer (and for the data evaluation of mobile station data against Svalbard

Lufthavn and Narveneset) and underlines the importance of the underlying surface (here marine terminating glaciers) on the footprint. Furthermore, spatial temperature variability as documented here will naturally have an impact on the temperature

comparison statistics between the matched pairs of mobile stations on Isfjorden, which do feature a slightly poorer match than the comparison for when the ships were all in the harbor, as pointed out above.

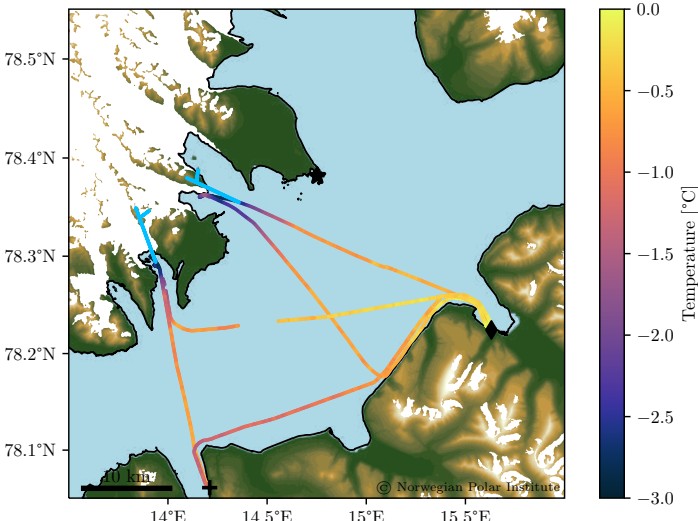

**Figure 14.** Near-surface air temperature anomalies (colored, solid line) based on observations from the IWIN mobile weather stations on MS Bard/Berg and MS Polargirl over Isfjorden using observations from the Bohemanneset lighthouse as reference (ship - lighthouse observations). The data are obtained during the time 06:00 – 17:00 UTC on 05 August 2022. Data from a stretch of the ship track in the middle of Isfjorden is missing due to a lack of GPS fix, as can be seen as a discontinuity in the ship track and corresponding temperature anomaly. In addition, wind observations from two example points are indicated by the wind barbs showing wind emanating from the glaciers in the northwest. The locations of Longyearbyen, Bohemanneset and Barentsburg are marked with a diamond, a star and a plus symbol, respectively.

    The fourth example is from 20 September 2022, during which Svalbard was dominated by moderate to strong synoptic flow, set up by a low pressure system over the Fram Strait to the west of Svalbard. This flow induced south – easterly winds over the Isfjorden region. For reference, the average wind speed measured at Bohemanneset during 06:00 - 17:00 UTC was

15.9 m/s. The observations from the mobile stations during this day reveal strong spatial variability in the wind field, both in terms of speed and direction (see Figure 15 (b)). The highest wind speeds are generally found over and downstream of Sassenfjorden. Here, the wind speed reached up to 20 m/s, which is more than twice the average of all wind speeds measured onboard MS Bard/Berg and MS Polargirl this day (8 m/s). The wind direction has a clear alignment with the surrounding topography of Sassenfjorden, i.e. from an easterly direction, in contrast to the synoptic flow which was more southerly on

this day. The relatively strong wind speeds and clear alignment with surrounding topography are indicative of so-called channeling (Whiteman and Doran, 1993). In contrast, very calm conditions with wind speeds of less than 2.5 m/s are found in mostly sheltered areas along steep coastlines, approximately perpendicular to the large-scale winds, e.g. in inner Billefjorden or along the coastlines north and west of Longyearbyen. Depending on the area considered and its respective topography, these differences occur over distances in the order of 100 m – 1 km. As already argued and shown through the first example above,

this variability in the atmospheric near-surface variables may reach levels similar to or larger than the values of the comparison statistics presented in Section 4. In turn, this means that the IWIN dataset and especially the measurements from the mobile stations have a large potential to investigate localized phenomena like channeling effects, drainage flows or differences in the surface energy balance over land and sea, and subsequent gradients in the near-surface atmospheric variables, especially temperature.

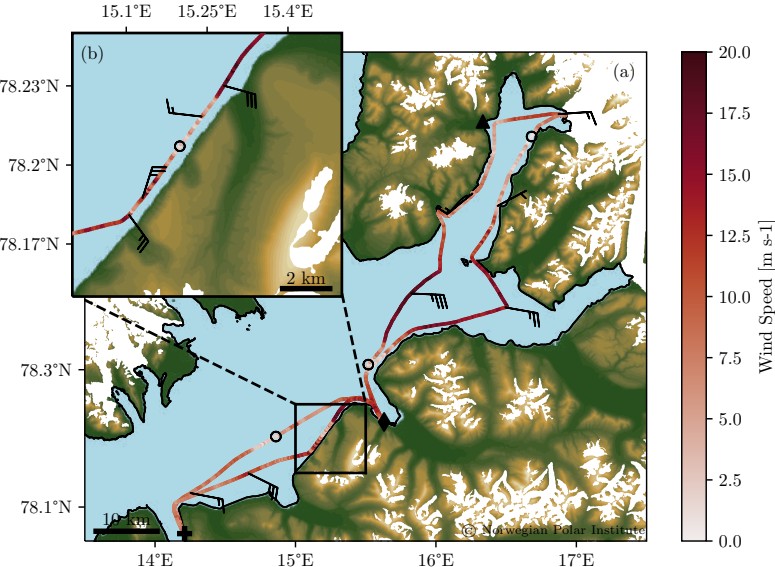

**Figure 15.** (a) Wind speed (solid line, red shading) and direction (wind barbs) over Isfjorden. (b) A zoom into an area along the coast line west of Longyearbyen with strong temporal and spatial variability in the observed wind. The data presented here are obtained from the IWIN mobile weather stations on MS Bard/Berg and MS Polargirl during 06:00 - 17:00 UTC on 20 September 2022. The locations of Longyearbyen, Pyramiden and Barentsburg are marked with a diamond, a triangle and a plus symbol, respectively.

# 6 Summary

IWIN is a new network of weather stations installed on lighthouses and ships in the Isfjorden region. The network is developed by UNIS with support from MET Norway and provides online, freely available and near real-time in-situ meteorological observations from large parts of the Isfjorden fjord system through MET Norway's data portals.

As a high-arctic fjord system, Isfjorden has been subject to strong climate change during the last couple of decades. It has gone from being dominated by Arctic Water during winter to being more dominated by Atlantic Water. Among the consequences is a rapidly diminishing sea ice cover. Atmospheric forcing plays a key role in this regard, which has been documented by among others Cottier et al. (2007); Muckenhuber et al. (2016); Nilsen et al. (2016); Skogseth et al. (2020). So far, these kinds of air-sea-ice studies have mainly relied on model and satellite data and sparsely available in-situ atmospheric observations. IWIN helps fill an observational gap over Isfjorden and it will thereby provide an important basis for future process studies of the importance of intra-fjord, local atmospheric forcing in this climatically important air-sea-ice interaction context (Stenlund, 2022). Indeed, the set of cases presented in this paper demonstrates the unprecedented (for the region) capabilities of IWIN to capture local- to meso-scale atmospheric phenomena. These phenomena include sharp contrasts in temperature and humidity between marine and terrestrial-based air masses, as evidenced by IWIN observations along the Isfjorden coast line. The observations also give insight to topographically induced wind phenomena during moderate to strong synoptic flow, such as channelling (fjord jet) and wake effects downstream of and along steep, coastal topography.

Furthermore, observations from IWIN address the need for improving our weather and climate prediction capabilities in the Arctic, which are not as good as at lower latitudes (Køltzow et al., 2021). IWIN does so by providing additional observations from an otherwise data sparse region, both for better process-level understanding (and thereby model and parameterisation development) and also for data assimilation purposes. Indeed, a recent master study at UNIS applying the IWIN observations in evaluating the AROME-Arctic weather model, run operationally by MET Norway for the Svalbard region, have revealed shortcomings (biases) in this system that likely otherwise would not have been detected (Schalamon, 2022). Also, IWIN will support the development and validation of new and ongoing efforts (see e.g. Valkonen et al. (2020)) for building capacity for hectometric-scale numerical weather simulations for the Arctic.

The Arctic is seeing a rather sharp increase in human activity in the form of among others tourism, shipping and research activity. Svalbard, and in particular Isfjorden, is no exception. IWIN helps enhancing the safety of such activity by providing online, freely accessible and near real-time weather data from Isfjorden.

In summary, IWIN supports our needs for a) better understanding and documenting local meteorological processes, relevant for and related to the ongoing rapid climate change in Svalbard, b) enhanced weather prediction capabilities through making unique in-situ observations available for model validation and assimilation purposes and c) enhanced safety for the increasing human activity in the region in the form of among others tourism, research and search and rescue missions through the provision of data that are freely available online in near real-time.

## 7 Outlook

In concert with the growing demand for in-situ observations in the climate-sensitive region of Svalbard, IWIN is under continuous development. Funding has already been granted for one additional lighthouse station (which will be installed at Kapp
Thordsen, 78.45632°N, 15.46768°E, during summer 2023) and one more mobile station. This fourth mobile station will be mounted onboard the UNIS RV Hanna Resvoll. In addition to the standard meteorological variables T, Q, WS, WD and p, it will also provide measurements of photosynthetically active radiation (PAR), which is critical for primary production both on sea and on land. Depending on the scientific needs of the researchers using using the ship, RV Hanna Resvoll will travel into parts of the fjord system not visited by the tourist cruise ships. In that way, the observations from the new station will nicely
complement those from the established stations presented in this paper.

Besides the expansion of IWIN by adding more stations, we continuously work towards assuring the highest-possible data quality, especially for the correction of the wind measurements from the mobile stations. One example are efforts currently underway to include satellite-compasses in the mobile station setups. These will give highly accurate estimates of the ships' motion and heading at any time and allow for more sophisticated wind measurement corrections, also when the ships are not
or only slowly moving.

## 8 Code and data availability

The IWIN data product described here (data until June 2023) can be found at Zenodo (https://doi.org/10.5281/zenodo.8137588). The complete IWIN dataset is available from MET Norway's THREDDS server (https://thredds.met.no/thredds/unis-obs/unis-obs.html), with the latest data becoming available in near-realtime (approximately 5 – 10 min time lag). THREDDS
is an open-source software solution for providing a way to publish and access scientific data in a distributed environment. It supports a wide range of remote data access, including OPeNDAP (Open-source Project for a Network Data Access Protocol).

Additionally, the full data set is registered at MET Norway's Arctic Data Centre (ADC, https://doi.org/10.21343/ebrw-w846, Frank et al. (2023)). The ADC offers additional functionality to access and download the dataset as ASCII-formatted CSV text files as well as direct visualization via a graphical user interface. This can be achieved by starting from https://adc.met.no/
metsis/search?fulltext=IWIN and selecting the dataset of interest via the "Child data"-buttons in the "Data access"-panels. In the end, the datasets can be downloaded or directly visualized using the respective buttons.

The data from MET Norway's reference surface weather stations in the Isfjorden region used for validation of the IWIN dataset is also available from MET Norway's data portals (e.g. https://seklima.met.no).

The Python and Matlab code used to process the data, produce the final data files and create the figures for this paper can be
found at https://github.com/lfrankunis/Iwin.

*Author contributions.* Lukas Frank is responsible for the publication of the dataset, including the processing of the raw data into netCDF files and the data transfer to MET Norway. He is also the corresponding author of this paper. Marius O. Jonassen is the main responsible for

the funding acquisition and has also contributed with data analysis and writing of this paper. Teresa Remes is responsible for the organisation of the data at MET Norway's servers. Florina Schalamon and Agnes Stenlund contributed with written text and figures in Sections 2 and 5.

*Competing interests.* The authors declare that they have no conflict of interest.

*Acknowledgements.* IWIN has been financially supported by a number of different institutions and projects, including UNIS, MET Norway, Alertness (NFR-280573), N-FORCES (NFR-337229) and the Jan Christensen foundation. These shall hereby all be acknowledged for their support. Stefan Claes provided invaluable help with the installation of especially the lighthouse stations and the coordination with Kystverket, which gladly allowed us to use the lighthouses as platforms and power sources. Special thanks to Henningsen Transport and Guiding, Polar-
charter, Hurtigruten and Brim Explorer, including their respective crews onboard the ships, for providing us with the mobile measurement platforms. Thanks also to Charlotte Gausa for her help with the installation of the stations and maintenance work.

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
