# Peer review of "IWIN: The Isfjorden Weather Information Network"

_Earth System Science Data, 2023_

## Referee Comment (RC2)

ESSD-2023-99, IWIN

Interesting system. I applaud authors intent and efforts to make (and, keep) data free and open. Also, as hinted, IWIN effort might encourage replication in other settings; without these kinds of additional deployments observational meteorological networks will increasingly fail to keep up with model (and data assimilation) resolution.

Unfortunately, description as presented here seems not quite ready for publication in ESSD. Please can authors read ESSD guidelines (short) at: https://www.earth-syst-sci-data.net/10/2275/2018/. Authors can of course withdraw manuscript to try a different journal. If they prefer to stay with ESSD, I request several changes.

1) Data access via THREDDS server (often-used standard for met data) and then OpenDAP (or proprietary software such as MATLAB or IDL) works for people already familiar with met data but does not meet needs of majority of ESSD readers. Because authors will want to preserve up-to-date access via MetNo servers, please may I suggest following strategy: capture a fixed-location fixed-time-period snapshot, as netCDF or even .csv, specific to this particular description, and provide same as a single download from MetNo or third-party repository (e.g. Zenodo or authors' preference). Do not make readers select coordinates or time periods; give us your best defined product. Data access 'statement' would then say something like: get data product described here at DOIxxx, while interested users can find most-recent and future data at DOIyyy (as for current IWIN data). Many ESSD papers use this distinction: snapshot here, real-time updates and future products at this other location. (To keep a two-year snapshot to reasonable download size, authors may need to revise time resolution, e.g. 10 min rather than 1 minute. Authors failed to justify either temporal resolution so choice should prove relatively easy.)

2) Uncertainties section seems incomplete. As authors imply, full uncertainty starts from sensors, includes deployment logistics, must accommodate failures in power or communication, and - finally - must exist within comprehensive metadata system (what sensor, where, for what purposes, on what platform, with what performance, what biases or drifts, what possible outliers, etc.). Authors make small effort (replaying info from manufacturer's data sheets) at sensor level but seem to then relax to standard MetNo procedures. Even where they correctly list some of these factors (e.g. ship plumes while - apparently - ignoring ship- or land-based elevation factors) they fail to prove understanding of relevant factors. Substantial literature exists, for example, on both ship-based and land-based deployment, footprint, and interference factors. Authors show no evidence that they recognize those factors. An unpublished MS thesis report does not substitute for careful thoughtful analyses. Because neither Campbell nor Gill manufacture their own temperature or RH sensors (do they use identical sensors?, doubtful), readers will have no basis to credit values cited in Table 3. I doubt that standard MetNo temperature sensors achieve resolution of 0.1K. Accuracy of 0.3K at 20C seems reasonable for laboratory, but not for median Svalbard temperatures. This reader suspects 20C represents almost absolute max for Longyearbyen. MetNo data show averaged daily means above 0C only for three or four summer months with annual average (even if rapidly warming) substantially below 0C. Many temperature and RH sensors perform poorly at -20C and fail at -40C. Why do authors then quote uncertainties based around +20C? As presented, this reader confronts a lack of useful comprehensive uncertainty quantification that would allow confidence in IWIN data.

3) Validation, absent. Authors refer to IWIN data as validation for, e.g., forecast models, or as an element of future improvements, but provide no validation of current IWIN data. If they intend their four examples as validation examples, they have missed for this reader. What ranges of T or RH in what wind speeds and directions? Reader finds no inter-comparisons with e.g. the four MetNo stations on southern shores of the fjord. MetNo has - in some cases - many decades of data from those stations; why does reader never see how IWIN data compare. Show how IWIN data fit within current MetNo or ECMWF forecasts? When

validating against openly-accessible (e.g. MetNo, ECMWF) data, include those data or a link to those data? Norwegian search and rescue already operates a local water temperature and wind product; how do IWIN data fit or not fit? Authors allude to inner fjord vs more exposed locations: show us how IWIN locations respond to cold meltwaters along a glacial front or to open warmer Atlantic water currents. Fundamentally, user needs abundant validation to develop trust in IWIN data. In current manuscript, they get none. For this reader, rather than four different map-based examples, I would rather see (and use, and learn to trust) two well-documented careful data inter-comparisons, one inshore and one offshore.

Potentially a useful ESSD product. Needs work, however. Authors to decide magnitude of the effort and their best options.

---

## Author Comment (AC1)

**Response to Reviewers**

Dear reviewers,

Thank you very much for your input on our manuscript "IWIN: The Isfjorden Weather Information Network". Below our collected responses to your individual comments:

**CEC 1, Ken Mankoff, 25 Apr 2023:**

⇒ Thank you for your very valid criticism and reference to the ESSD Editorial Guidelines. We hope our modifications of the manuscript now meet the requirements of your journal.

I am concerned with how this manuscript fits within ESSD. Per the paper itself, it is not a data description paper. From L49, "The primary goal of this paper is to provide documentation on the instrumental setup of IWIN." The current paper does not describe the data. Further out-of-scope, it uses the data to describe various weather events from page 10 to 15, something we normally discourage.

⇒ In order to shift the focus of the manuscript to the (IWIN) data set itself, we have added detailed information on the measurements and the subsequent processing steps from the raw measurements to the published data files (Section 3). Furthermore, we now provide a new overview figure of the time series measured by the lighthouse stations (Figure 10). We have re-written large parts of Section 5 that previously showcased various weather events. The section is now cast such that it presents examples of natural (spatial and temporal) variability in the data set and how this variability affects the evaluation statistics (see also comment and response below).

I think with little effort this paper could be published in another journal - withdraw is an option. If it stays in ESSD and meets the ESSD Editorial Guidelines https://essd.copernicus.org/articles/10/2275/2018/ it would take quite a bit of work. I encourage the authors to do that work.

There is currently *no validation*. ESSD expects *extensive* validation. Are there weather stations nearby that can be used? If you turned each of the weather examples into a validation, that would be impressive, and a lot of work. You could do fine with half of the examples.

⇒ We would still like to publish our manuscript with ESSD and we have now added extensive validation/evaluation of the IWIN data set, comprising a whole new section (Section 4). The section contains an intercomparison of data from the different IWIN stations (mobile and lighthouse) as well as a comparison to reference stations from MET Norway in the area. Furthermore, we use the example weather events presented in Section 5 to evaluate the so-inferred validation statistics with respect to the natural variability present in the region and its impact on the evaluation statistics.

I'm also uncertain with how certain you are of your small uncertainties. Are there really no additional uncertainties beyond the stated manufacturer uncertainties? Are you deploying these in the expected settings and environment? A ship structure may (or may not) not influence the results, but what about a bright red lighthouse roof? Also, I question citing a masters thesis for the 'no error from ship' claim. That thesis appears to focus on wind direction, as does your almost entire paragraph of 'uncertainty'. What about T? RH? P? Fig 6.2 of that masters thesis appears to show *significant* disturbance in streamline plots - do these impact P or T? I would presume so.

⇒ As a result of the added, in-depth validation of the IWIN data set in the new version of the manuscript, we now have a considerably better grasp of the remaining uncertainties and where they originate from and corresponding text has been added to the validation section (Section 4). In addition, an improved discussion of remaining uncertainties is presented following the validation in Section 4. As outlined in that section, we are confident in the quality of the T, RH, P and WS measurements as underlined by the (very) favourable validation statistics for these variables. See for example the internal comparison of the mobile station data when the respective ships (with rather different physical characteristics) were all located in the Longyearbyen harbor (Table 5). Comparisons between stations that are farther apart (up to 2km, as in the comparison of the mobile station data with Svalbard Airport and the two IWIN lighthouses), reveal larger deviations between the respective data sets and variables. We argue that these deviations in large can be explained by natural, atmospheric variability. As now also described in the new version of the manuscript, we are less confident in the wind direction data, and we offer several explanations for this and future steps for further improving measurements and estimations of this variable (including the installation of a new GPS-based compass for better estimation of the ships' heading). Future work will also include a more in-depth analysis (and quality check) of the CFD simulations. Regarding the bright red roof of the lighthouses, it is hard to directly estimate the impact of this as we currently lack "true" reference data from these locations. However, considering the good agreement between all different lighthouse stations as well as mobile and lighthouse stations during times a ship is in close proximity of the respective lighthouse as well as the absence of unnaturally high peaks in e.g. the temperature measurements around noon (when a potential impact of solar radiation can be assumed to be largest), we believe the effect is negligible compared to the other uncertainties previously discussed.

Finally, in addition to describing the data product, it would be good to describe the data files. What format? What metadata? What are your CSV columns? How are the data processed? Where is the software that did this processing?

⇒ As already indicated above, the updated manuscript now includes a step-by-step description of the data processing from the raw measurements to the final data files. Furthermore, we provide additional information on the metadata stored in the published data files. In the data availability section, we give detailed instructions on where and how the data can be accessed, including online visualization and options for the download of .csv files using a GUI. All software used to process the data (and produce the figures for this manuscript) is now publicly available on Github (https://github.com/lfrankunis/Iwin).

**RC1, Anonymous Referee #1, 12 May 2023:**

This manuscript describes a new weather observation network in Svalbard as well as some examples of local meteorological features observed by the network. The manuscript is well written and largely provides all relevant information about the network. I offer a few minor comments / suggestions below but otherwise find this manuscript suitable for publication after minor revisions.

⇒ Thank you for this encouraging and motivating feedback. We have updated our manuscript according to your suggestions and hope we meet your expectations.

Minor comments

Mention content of section 3 in last paragraph of section 1.

⇒ We have added links to the data in Abstract and Introduction.

Table 1: In addition to the measurement altitude above sea level please include the height above ground level for each lighthouse station.

⇒ As all lighthouses are the same height, we have opted for providing this information once for all stations in the main text, instead of repeating the same values in the table.

Table 2: Provide station height above sea level for each ship-mounted station.

⇒ Similar to the previous comment, this height is similar for all mobile stations and therefore given in the main text.

Section 3: Is icing a concern and what impact does it have on the sensors and measurement errors?

⇒ Due to the remote locations the stations are installed, we are not able to visually verify that icing is not occurring. However, we do not see any sign of sensor malfunctioning in the data reminiscent of icing issues. We have updated the manuscript with corresponding information (Section 4.2.4).

It would be useful to give a brief summary of how the measurements are made. For example, are the wind measurements made via a sonic approach?

⇒ The manuscript has been updated with detailed information on the whole chain starting from the raw measurements via the processing steps and to the final published data files, including the principle employed by the sensors to measure wind (which is indeed based on a sonic approach).

Line 158-160: I was unable to find a way to download or visualize the IWIN data from the link provided in this dataset. Please provide additional details about how a user can easily access and visualize this data.

⇒ We have moved the data availability section to the end of the manuscript and added more detailed instructions on how to find the download and visualization tools at the ADC webpage.

Line 171: Figure referenced here should be 10a not 3a. On line 175 Figure reference should be 10b not 3b.

⇒ Thanks for this well-spotted mistake, we have double-checked all figure and table references now.

**RC2, Anonymous Referee #2, 28 May 2023:**

Interesting system. I applaud authors intent and efforts to make (and, keep) data free and open. Also, as hinted, IWIN effort might encourage replication in other settings; without these kinds of additional deployments observational meteorological networks will increasingly fail to keep up with model (and data assimilation) resolution.

⇒ Thank you for your recognition of our efforts and your valuable input to improve our manuscript.

Unfortunately, description as presented here seems not quite ready for publication in ESSD. Please can authors read ESSD guidelines (short) at: https://www.earth-syst-sci-data.net/10/2275/2018/. Authors can of course withdraw manuscript to try a different journal. If they prefer to stay with ESSD, I request several changes.

⇒ We have substantially changed our manuscript to adjust for the requirements specified in the guidelines. We hope that our manuscript in its new form now meets these requirements.

1) Data access via THREDDS server (often-used standard for met data) and then OpenDAP (or proprietary software such as MATLAB or IDL) works for people already familiar with met data but does not meet needs of majority of ESSD readers. Because authors will want to preserve up-to-date access via MetNo servers, please may I suggest following strategy: capture a fixed-location fixed-time-period snapshot, as netCDF or even .csv, specific to this particular description, and provide same as a single download from MetNo or thirdparty repository (e.g. Zenodo or authors' preference). Do not make readers select coordinates or time periods; give us your best defined product. Data access 'statement' would then say something like: get data product described here at DOIxxx, while interested users can find most-recent and future data at DOIyyy (as for current IWIN data). Many ESSD papers use this distinction: snapshot here, real-time updates and future products at this other location. (To keep a two-year snapshot to reasonable download size, authors may need to revise

time resolution, e.g. 10 min rather than 1 minute. Authors failed to justify either temporal resolution so choice should prove relatively easy.)

⇒ To the authors, this comment came as a bit of a surprise. As pointed out by the reviewer, storing meteorological (and similar spatio-temporal data e.g. from oceanographic observations, climate models etc.) as netCDF files is, to our knowledge, considered the community standard and easiest way of complying to FAIR principles and conventions such as CF or ACDD, as all meta data can be stored together with the measurements in the same file. By publishing the data via MET Norway's data portals, it is ensured that these FAIR principles are followed, and the data can be accessed using both programming solutions or a GUI. Zenodo offers only very limited options to find a dataset, if the respective DOI is unknown (no options for filtering a search e.g. for a region, time period, etc.). In order to avoid publishing the same data twice at different places, we hope to be allowed to stick to our current setup. We would also like to point out that users, also not familiar with meteorological data in general, can easily access the data via the GUI at the adc.met.no website hosted by MET Norway. As we (now) also describe in the new version of the manuscript this website allows users to download the IWIN data in the form of .csv files, which should be more accessible to the general audience as they can be easily opened in for example Excel. This is in contrast to netcdf files (that typically requires specialised software for reading). On adc.met.no one can also easily visualise data time series and one can also select (zoom in on) desired time periods and thus view snapshots of selected time periods. It is also possible via adc.met.no to select the desired data time resolution (e.g. 1 or 10 min).

2) Uncertainties section seems incomplete. As authors imply, full uncertainty starts from sensors, includes deployment logistics, must accommodate failures in power or communication, and - finally - must exist within comprehensive metadata system (what sensor, where, for what purposes, on what platform, with what performance, what biases or drifts, what possible outliers, etc.). Authors make small effort (replaying info from manufacturer's data sheets) at sensor level but seem to then relax to standard MetNo procedures. Even where they correctly list some of these factors (e.g. ship plumes while - apparently - ignoring ship- or land-based elevation factors) they fail to prove understanding of relevant factors. Substantial literature exists, for example, on both ship-based and landbased deployment, footprint, and interference factors. Authors show no evidence that they recognize those factors. An unpublished MS thesis report does not substitute for careful thoughtful analyses. Because neither Campbell nor Gill manufacture their own temperature or RH sensors (do they use identical sensors?, doubtful), readers will have no basis to credit values cited in Table 3. I doubt that standard MetNo temperature sensors achieve resolution of 0.1K. Accuracy of 0.3K at 20C seems reasonable for laboratory, but not for median Svalbard temperatures. This reader suspects 20C represents almost absolute max for Longyearbyen. MetNo data show averaged daily means above 0C only for three or four summer months with annual average (even if rapidly warming) substantially below 0C. Many temperature and RH sensors perform poorly at -20C and fail at -40C. Why do authors then quote uncertainties based around +20C? As presented, this reader confronts a lack of useful comprehensive uncertainty quantification that would allow confidence in IWIN data.

⇒ In order to give the reader a better understanding of how the IWIN data are handled from the raw measurements to the final published data files, we have included detailed information on the different processing steps, starting from information on the actual sensors and measurement principles via the software used to process the data to the final metadata included in the published data files. Furthermore, the newly-added Section 4 on validation of the dataset (see also following comment and respective reply) enables us to better estimate remaining uncertainties and their potential sources. By using the examples presented in Section 5, we are furthermore able to attribute large portions of the remaining

uncertainties to natural, atmospheric variability in the measured atmospheric variables, which is expected in the region and one of the main reasons to establish this new weather station network in the first place. Regarding the operating temperatures, the manufacturers' specifications might be given for laboratory conditions at +20degC, however, from the data record we see that minimum temperatures rarely fall below -20degC, which is well within the operating range. We furthermore do not see any signs of deteriorating performance compared to MET Norways surface reference stations during extreme weather events such as very low temperatures or high wind speeds.

3) Validation, absent. Authors refer to IWIN data as validation for, e.g., forecast models, or as an element of future improvements, but provide no validation of current IWIN data. If they intend their four examples as validation examples, they have missed for this reader. What ranges of T or RH in what wind speeds and directions? Reader finds no inter-comparisons with e.g. the four MetNo stations on southern shores of the fjord. MetNo has - in some cases - many decades of data from those stations; why does reader never see how IWIN data compare. Show how IWIN data fit within current MetNo or ECMWF forecasts? When validating against openly-accessible (e.g. MetNo, ECMWF) data, include those data or a link to those data? Norwegian search and rescue already operates a local water temperature and wind product; how do IWIN data fit or not fit? Authors allude to inner fjord vs more exposed locations: show us how IWIN locations respond to cold meltwaters along a glacial front or to open warmer Atlantic water currents. Fundamentally, user needs abundant validation to develop trust in IWIN data. In current manuscript, they get none. For this reader, rather than four different map-based examples, I would rather see (and use, and learn to trust) two well-documented careful data inter-comparisons, one inshore and one offshore.

⇒ We have added extensive validation material on the IWIN dataset in the new version of the manuscript and dedicated a new section to this (Section 4). In our opinion, the results are very favourable and speak for the quality of the IWIN data set. Some uncertainties remain, though, especially with regards to wind direction from the mobile stations, which we have added a discussion of in the manuscript along with suggestions and planned efforts for future improvements. Other uncertainties (reasons for deviations) inherent in the comparison/validation statistics include natural, atmospheric variability, and we now use the presented examples/cases (of weather phenomena) to support our discussion around this. In these examples, we also allude to how the IWIN data set gives (unprecedented for the area, to our knowledge) insight to for example the (terrestrial) footprint of marine terminating glaciers on the adjacent near-surface atmosphere (temperature, wind etc) over the fjord. In addition, in this new Section 4 we put the lighthouse station data in context of data from permanent MET Norway reference stations. We describe how the location of the western-most station (the IWIN station at Daudmannsodden lighthouse) indeed is dominated (warmer) by the open, Atlantic ocean in the west and that the Narveneset lighthouse data are in contrast more "continental" (colder especially during winter). The (newly added) validation includes comparisons between different IWIN stations (mobile vs mobile, mobile vs lighthouse stations etc) as well as to surface reference stations from MET Norway and data from the research vessel Kronprins Haakon. In the absence of any reference observation data, such as these, we agree that it would be relevant to compare the IWIN data to model data such as from MET Norway and ECMWF. However, given the presence of said reference station data we believe that comparing the IWIN data against those is far more relevant than comparing against model data. These model data are very likely a fair bit more uncertain than the observations, including the IWIN data themselves.

Potentially a useful ESSD product. Needs work, however. Authors to decide magnitude of the effort and their best options.

On behalf of all co-authors,

Best regards,
Lukas Frank